# Towards a Better Understanding of the Complexities of Myalgic Encephalomyelitis/Chronic Fatigue Syndrome and Long COVID

**DOI:** 10.3390/ijms24065124

**Published:** 2023-03-07

**Authors:** Warren P. Tate, Max O. M. Walker, Katie Peppercorn, Anna L. H. Blair, Christina D. Edgar

**Affiliations:** Department of Biochemistry, School of Biomedical Sciences, Division of Health Sciences, University of Otago, Dunedin 9054, New Zealand

**Keywords:** ME/CFS, Long COVID, systemic inflammation, neuroinflammation, disease subtype/phenotype, susceptibility, disease models, therapeutics

## Abstract

Myalgic Encephalomyelitis/Chronic Fatigue Syndrome (ME/CFS) is a complex condition arising in susceptible people, predominantly following viral infection, but also other stressful events. The susceptibility factors discussed here are both genetic and environmental although not well understood. While the dysfunctional physiology in ME/CFS is becoming clearer, understanding has been hampered by different combinations of symptoms in each affected person. A common core set of mainly neurological symptoms forms the modern clinical case definition, in the absence of an accessible molecular diagnostic test. This landscape has prompted interest in whether ME/CFS patients can be classified into a particular phenotype/subtype that might assist better management of their illness and suggest preferred therapeutic options. Currently, the same promising drugs, nutraceuticals, or behavioral therapies available can be beneficial, have no effect, or be detrimental to each individual patient. We have shown that individuals with the same disease profile exhibit unique molecular changes and physiological responses to stress, exercise and even vaccination. Key features of ME/CFS discussed here are the possible mechanisms determining the shift of an immune/inflammatory response from transient to chronic in ME/CFS, and how the brain and CNS manifests the neurological symptoms, likely with activation of its specific immune system and resulting neuroinflammation. The many cases of the post viral ME/CFS-like condition, Long COVID, following SARS-CoV-2 infection, and the intense research interest and investment in understanding this condition, provide exciting opportunities for the development of new therapeutics that will benefit ME/CFS patients.

## 1. Background

Myalgic Encephalomyelitis/Chronic Fatigue Syndrome (ME/CFS) is a neurological disease with strong immune/inflammatory features and with a pathophysiology not yet fully understood. ME/CFS is the umbrella name for a complex syndrome arising in people predominantly after a viral infection, but also initiated by other major stressors such as non-viral disease agents, major surgery, exposure to toxic agricultural chemicals or even simply severe stress [1]. It has proven to be a poorly understood disease and, for the majority of those affected, lifelong. Without an established molecular diagnostic test, a diagnosis is made if the core symptoms fitting a clinical case definition persist for six months [2]. By contrast, Long COVID encompasses people with a post-viral syndrome arising from one specific virus, SARS-CoV-2, but it also includes a group whose illness relates to organ damage from the virus infection [1]. Long COVID also differs from ME/CFS with regards to the estimated number of cases and the speed in which cases have arisen. Long COVID cases have accumulated rapidly following the SARS-CoV-2 pandemic involving in excess of 650 million people whereas ME/CFS cases have gradually accumulated over decades following isolated infections. Nevertheless, the clinical case definition derived by the World Health Organization for Long COVID is very similar to that accepted now for ME/CFS. All comparative studies to date point to it being a classic post viral syndrome such as ME/CFS, perhaps with some specific features relating to the specific infectious virus [3,4].

A common initiator of ME/CFS is infection with the endemic virus, Epstein Barr Virus, that causes glandular fever, and yet it is estimated that perhaps only one in 10–20 develop the post-viral syndrome after being infected with this common virus [5]. With Long COVID, the reported percentage arising from infections with the SARS-CoV-2 virus has varied widely in different publications [4,6,7,8] and likely it is different with the later variants [4]. A very recent report identified the relative odds of Long COVID during the omicron period of infection vs that of the delta variant and found a reduction for omicron of 0.24–0.5 depending on age and time since vaccination [9]. Nevertheless, the worldwide health burden of up to 50 million with ME/CFS will be added to by perhaps 100 million with Long COVID.

The intense interest and research activity now being given to Long COVID has promise to accelerate our understanding of ME/CFS for the patient group and provide evidenced-based therapeutic options [10]. In 2021, Friedman and co-authors discussed the importance of classifying Long COVID, ME/CFS and other similar chronic conditions as Post-Active Phase of Infection Syndromes or PAPIS for the advancement of research and clinical care [11]. Currently, a topical collection in the MDPI publication *Healthcare* is dedicated to this concern [12]. The hope is that this new research will enable a reversal of the dysfunctional physiology in ME/CFS and Long COVID that is so widespread in affected patients.

## 2. Susceptibility to Developing ME/CFS or Long COVID

There is little understanding of what makes a person susceptible to developing these post-stressor diseases. A stressor can be a virus, another infectious agent, an environmental toxin, or a serious stress event in the life of the susceptible person. What are the contributions of prior health history or the exposures to priming events for those succumbing to the syndrome? Is susceptibility a genetic susceptibility, an environmental consequence, or a combination of both? Unlike the typical response, those susceptible to a post-stressor disorder do not simply mount a transient response to combat the external stress and quickly recover, but rather develop a significant chronic response that spirals into the ongoing pathophysiology of the fatigue syndromes. The body perceives there is continuing danger [13].

To investigate susceptibility factors in ME/CFS and Long COVID, we conducted a quantitative survey with 160 ME/CFS patients and 57 of the first wave of Long COVID patients in New Zealand to determine whether their prior personal history was a predictor of developing the syndrome, and whether family history was indicative of genetic factors being significant [14]. The phenotypic overlap found between Long COVID and ME/CFS participants in this study in terms of symptomology, severity of symptoms and capacity for activity provided further support for the suggestion that Long COVID and ME/CFS are closely related conditions as hypothesised by published literature [15,16,17,18,19], albeit perhaps with certain features specific to the SARS-CoV-2 virus.

The initiating triggers of ME/CFS in patients and the severity of the SARS-CoV-2 infection in the case of the Long COVID patients is shown in Table 1A. Only 16.6% of participants reported having no underlying health conditions, whereas many patients reported preexisting underlying health conditions (Table 1B). For conditions not specified in the questionnaire, reported as ‘other’ (45.6%), the most common were asthma, endometriosis, Ehlers–Danlos Syndrome and fibromyalgia. Of the specified conditions, gastrointestinal issues were the most reported condition (42.4%). Mental health conditions were more commonly reported in ME/CFS (36.3%) than in Long COVID (17.5%). Moreover, many reported having experienced relatively long times to recover from childhood illnesses (70% of ME/CFS patients, and 45% of Long COVID patients took between two weeks and two months). ME/CFS patients reported childhood illnesses were frequent (28.5%), while those with Long COVID somewhat less (17.5%). Hence, there was an overall pattern of a history of frequent illness during childhood that required significant time to recover, followed by underlying health conditions prior to contracting ME/CFS or Long COVID from the initiating stress. The environmental effect might be from a vaccination [20], a previous viral infection [21] or stress from a life crisis [22]. Either singly (genetic or environmental) or in combination, such effects might provide an impetus in a susceptible person for the health response to the stressor not being transient as in most people, but for it to become chronic and ongoing. If that extends to more than six months, then the criterion for a diagnosis of ME/CFS is met.

Was there any suggestion of a genetic susceptibility in the patient cohorts? There was a significant relationship between the participants’ ongoing fatigue illnesses and having a family member with similar symptoms (Fisher’s exact test gave a *p*-value of *p* = 0.019). Additionally, the relationship between the participants’ fatigue illnesses (i.e., ME/CFS or Long COVID) and having a family member who developed Long COVID after an initial COVID-19 infection gave a *p*-value with the Fisher’s exact test of *p* ≤ 0.001. This indicated a likely genetic component.

Genome-wide association (GWAS) studies with relatively small sample sizes, aimed at identifying genetic features linked to developing ME/CFS, identified, at best, possible genetic risk loci [23,24,25]. Larger numbers were examined in three ME/CFS cohorts: a Norwegian discovery cohort (*n* = 427), a Danish replication cohort (N = 460) and a replication dataset from the UK biobank (N = 2105) incorporating 2532 patients for the genome-wide analyses and 460 patients for a targeted analysis. Even with this enhanced study, ME/CFS risk loci displaying genome-wide significance were not identified [26]. However, very recently, compelling support for genetic susceptibility came from a combinatorial analysis of clusters of loci from a genome-wide association study. It identified clusters of 199 single-nucleotide polymorphisms connected to 14 genes that could identify 91% of patients from the UK biobank selected for the study [27]. These genes could be linked to processes likely dysfunctional in ME/CFS. This is the strongest evidence yet that there are susceptible groups in the population who, because of their genetic profiles, are at risk of developing these diseases when they are subject to an external stressor, whether it be viral infection or another stress event.

Collectively, our quantitative analyses and the recent ME/CFS GWAS/combinatorial study suggest that both ‘nature’ in the form of the genetics of the individual and ‘nurture’ from the impact of the environment on their personal health history might together determine the susceptibility to whether a patient develops ME/CFS or Long COVID following exposure to a significant stressor. This is indicated in Figure 1.

A susceptible genetic profile leads to a physiology that predisposes an individual at risk of developing a chronic immune/inflammatory response when there is a new environmental trigger. This risk might be further enhanced by a health history of exposure to viruses or particularly stressful life events. That history alone may be sufficient to provide sufficient susceptibility for transition to a chronic immune response when exposed to the new trigger event that eventually leads to the neurological symptoms of ME/CFS.

## 3. What Is the Significance of ‘Subtypes/Phenotypes’ in ME/CFS?

The concept of subtypes

There are at least a hundred symptoms ascribed to ME/CFS and reported by those affected, but most are not found in all patients. A smaller number are shared by most ME/CFS patients, and these are used in diagnosis. Many research publications have concluded there are subtypes within the ME/CFS population following the emergence of different patterns in patient studies, but understandably it has often been difficult to define what the term subtype really means in the context of the study. Previously variable patterns of symptoms were explained by some patients being misdiagnosed with ME/CFS, however this would now seem to explain only a small component of any differences in disease course and severity. Clinical case definitions for ME/CFS have since become well known and more refined [28,29], so targeting to each patient considering their specific phenotype/subtype aims to give them the most effective treatment.

Why might subtypes/phenotypes within the ME/CFS population exist? They might arise in ME/CFS because of the nature of the different initiators of the disease among the patient cohorts under study. The specific virus, or type of severe stressor, could have unique contributions to the development of a particular disease profile. There is a variety of known “trigger events” that have been linked to ME/CFS. ME/CFS most often develops following an infection by a pathogen such as Epstein Barr virus, Ross River virus, an enterovirus or following a major stress event such as surgery or exposure to toxins such as ones used in agriculture [30]. Such a wide array of trigger events causing ME/CFS suggests that there could be multiple phenotypes/subgroups within the ME/CFS population as a result. The symptomology seen in ME/CFS provides further evidence of this.

The ‘ME/CFS-like’ Long COVID patients suffering from the post viral fatigue syndrome have a single originating stressor, the SARS-CoV-2 virus. They provide a unique opportunity to evaluate whether subtypes might arise from different initiators [31]. Long COVID has some unique characteristics that apparently relate to the specific SARS-CoV-2 virus, such as the loss of taste and smell, and conditions such as enhanced skin problems, a loss of voice control, and breathlessness in some of those affected. However, most of the many symptoms ascribed to ME/CFS have also been associated with Long COVID (and are also found in only a proportion of the cases in this cohort). This would indicate there are multiple phenotypes among Long COVID patients as well, and indeed analysis of health record data from two large patient cohorts has identified four subtypes [32].

This suggests while each specific stressor trigger for ME/CFS may add some unique features to the pathophysiology of ME/CFS, it is likely not the major cause of the occurrence of the different phenotype/subtypes. It seems more likely that there is a continuous spectrum of pathophysiological responses within each individual because of their genetic profile/health history that determine their disease course.

Why might subtypes be important?

Subtypes can be useful for categorising clusters of patients who have similar physiological and neurological responses. This will be particularly important in the future when patients can be specifically targeted with individual treatments as they become available. Current experience is that some patients can show benefit from a treatment, others no change and yet a third group have their symptoms become worse. It is of interest that ME/CFS women show different responses as well to natural physiological changes such as pregnancy. Remarkably, some women have a significant amelioration of their symptoms and experience a year of relatively good health during pregnancy, whereas others experience no change in their ME/CFS disease profile, or have their condition become significantly worse as a result of the pregnancy. Hence, the same physiological changes cause quite disparate effects to suggest there are at least three subgroups—those that ‘improve’, have ‘no change’, or ‘relapse’. One study showed that 41% of women had no change in ME/CFS symptoms during pregnancy, while 30% showed an improvement in symptoms (went into near remission) and the remaining 29% had relapsing symptoms [33].

Medications

This same phenomenon of benefit/no change/harm is also found not uncommonly with promising medications or supplements that can help some patients while showing no benefit for others, and even cause an exacerbation of symptoms. The anti-neuroinflammatory drug Low-Dose Naltrexone (LDN), for example, currently a drug of choice to help ameliorate symptoms in ME/CFS patients, fits this profile, as indicated from patient/clinician histories. Clinical trials are now currently in progress to determine the effectiveness of LDN for the ME/CFS-related illness, fibromyalgia [34]. A Stanford trial evaluated a dopamine D2 agonist, aripiprazole, on 101 ME/CFS patients [35], since dopamine has been linked to regulation of immune cell function and neuroinflammation, prominent features of ME/CFS. Of the trial participants, 74% showed some improvement in the core defining symptoms of fatigue, brain fog, sleep quality and a reduction in post-exertional malaise, while 12% showed no change, and 14% had worsening symptoms. Antivirals can reduce fatigue and increase natural CD4+ T cells and natural killer cells, but only in a subgroup of ME/CFS patients [36]. These may be effective in patients that show evidence of viral reactivation.

One of the earliest descriptions of subtypes/phenotypes in ME/CFS was defined from gene expression profiles by Kerr and co-workers in 2008 [37]. They found differential expression of 88 human genes in patients with ME/CFS. Clustering of quantitative PCR (qPCR) data from patients revealed seven distinct subtypes with distinct differences in clinical phenotypes and severity. Jason and co-workers in 2018 suggested that subtyping patients with ME/CFS according to illness course is a promising method for creating more homogeneous groups of patients, with the documentation of significant symptomatic and functional differences between the groups [38]. In 2020 Nacul and coauthors highlighted the severity, progression, and duration differences among different individuals, and the changing way disease manifests itself in each individual with time lending weight toward the “categorization of different subtype trajectories of ME/CFS” that may differ in pathogenesis and prognosis [39]. In 2021, a Spanish cluster analysis within a ME/CFS study identified five subtypes of ME/CFS from clinical phenotypes coupled with multiple questionnaires assessing medical history, fatigue and pain, post-exertional malaise, immune, and neuroendocrine features, sleep quality, dysautonomia, cognitive function, anxiety and depression and functionality [31].

While these studies are informative, there is the obvious danger of individual researchers creating a plethora of different putative subtypes according to the disease features they are studying, rather than groupings that can usefully provide benefit for every patient. Nevertheless, characterising subtypes/phenotypes seems a useful way to help progress understanding and beneficial treatment strategies for ME/CFS patients. Categorising subtypes/phenotypes on the basis of those individuals who respond to specific treatments may allow treatment plans within a precision medicine framework to be created for each ME/CFS patient.

Recently, we carried out longitudinal studies on individual ME/CFS patients to evaluate whether they have molecular and physiological diversity or have very similar changes. We studied a small cohort of young women with ME/CFS of very similar age and lifestyle who have had similar severity, length and course of illness, as well as, importantly, a similar degree of functionality. We inferred in the context of a phenotype/subtype model for the disease course they would be classified within a single subtype—they all had a moderate but classic ME/CFS profile and limited functionality, diagnosed by the same clinical case definition and by the same expert ME/CFS clinician. These young women were examined individually in a comprehensive post-exertional malaise study along with age- and gender-matched controls. Over two standard exercise protocols, and in longitudinal molecular studies of their energy production and oxidative stress [1], each ME/CFS study subject showed unique characteristics in both their physiological and molecular responses. Figure 2A illustrates individual responses of three of the ME/CFS patients in their mitochondrial energy production, measured by a stress test with the Seahorse analyser in purified PBMC’s before exercise, and at 24 h and 48 h after the two exercise sessions that were at 0 h and 24 h. The control subject C036 showed no change in their mitochondrial profile of oxygen consumption in the stress test through the longitudinal study. By contrast there were markedly different responses among the three individual patients shown here; one showed no change, like the control, one showed a marked decrease by the 24 h time point and the other a decrease but not until 48 h. These are limited observations derived from within a larger study, but collectively, with the further observations shown in the separate studies of Figure 2B,C, they reinforce the conclusions that there is individual patient variation within a very similar phenotype of ME/CFS. Figure 2B likewise shows variability among the patient subjects for a biomarker of oxidative stress, in this case carbonylation modification of PBMC proteins. ME026, like the two controls, showed a decrease from 0 to 48 h, ME007 a significant increase, and ME024 and ME028 no change. Hence this study suggested ME/CFS patients who would be classified according to symptoms, and disease course and severity in the same phenotype/subtype can have quite different biochemical/physiological responses.

Furthermore, ME/CFS patients seem to respond differently at a detailed molecular level. In a separate year-long longitudinal DNA methylome study through a debilitating relapse and then recovery to pre-relapse health status [40], two of these ME/CFS subjects exhibited unique site-specific changes in their methylome during their relapse that were restored on recovery. Figure 2C shows three unique examples for each subject of individual bases on the genome that undergoes a change during the relapse (shaded areas). Patient 1 had a relapse of two months and patient 2 for only one month. These changes in DNA methylation then recovered to the original level. The overall functional changes deduced from all the changes identified in each subject are shown in Figure 2D with the individual variably methylated fragments (iVMFs), the genomic features with which they were associated, the associated numbers of genes, and the functional categories that were affected during the relapse. As shown in Figure 2D, the effects of the relapse on their physiology and molecular pathways were generally similar (immune, metabolism, mitochondrial energy production) but there were also changes specific to each ME/CFS patient.

The ME/CFS patient cohort described above and inferred to be from one subtype additionally had diverse responses to the RNA vaccine against SARS-CoV-2. The effects spanned from very mild, typical of healthy young women, to more severe but not long-lasting, but in one case the adverse reaction was serious enough to require hospitalisation post-vaccination. This suggested that while they exhibited a similar ME/CFS disease phenotype, their immune systems were responding quite differently to this reactive vaccine. A quantitative survey from the national ME/CFS disease advisory association, ANZMES, of the effects of the vaccine on ME/CFS patients in New Zealand recorded a high frequency (1 in 4) of New Zealand ME/CFS patients having ongoing severe adverse reactions to the vaccine (i.e., a major ME/CFS relapse) [42]. By contrast in this survey about 1 in 20 reported an improvement in their ME/CFS following vaccination. Such a high frequency of adverse reactions in ME/CFS patients is consistent with those reported by patient groups in the United States [43]. Interestingly, vaccination can be a trigger for developing ME/CFS in healthy people, for example, with the influenza vaccine. Now there are anecdotal reports from clinicians who have also seen the onset of ‘Long COVID’ or ME/CFS-like illness following vaccination with RNA vaccines (in New Zealand the BioNTech/Pfizer RNA vaccine has been almost exclusively used). Presumably, these affected people either carry a ‘silent’ genetic susceptibility that is challenged with the reactive vaccine, or their current state of personal health from previous environmental exposures to viruses, other microorganisms or toxic chemicals makes them at least transiently particularly vulnerable to reacting in this way to the vaccine. For those people with existing ME/CFS who showed a high susceptibility for a significant ongoing relapse after vaccination, it is assumed their chronically activated immune system is close to a critical threshold and the reactive vaccine is sufficient to drag the illness back into a more severe state.

In conclusion, although cases in ME/CFS can be usefully divided into subtypes based on their disease profiles, there are still diverse differences likely within members of each subtype that can manifest in different responses to typical day-to-day activities and stresses such as exercise and vaccination. This highlights the challenge of developing universal therapies for ME/CFS patients that will manage their illness better and enable higher functionality.

## 4. The Search for a Simple Universal Diagnostic Test for ME/CFS

Currently, ME/CFS has no conclusive molecular/cellular-based diagnostic laboratory test, and this gap has meant that patients and families even today are left often without the support of their healthcare system and social support systems because of an equivocation about their illness. When results of a typical blood screen show no clear abnormalities there is frustration for both the patient and the practitioner as to determining the best course of action.

It is heightened when, as sometimes occurs, severely debilitated patients are told after the blood screen with kind intention by their clinician “I have good news, there is nothing wrong with you”. Sadly, patients report feeling dismissed and abandoned from this response, with their debilitating illness not acknowledged. Despite clinical case definitions for diagnosis being refined considerably in recent times, a diagnostic molecular biomarker, tool, or accessible procedure specific for ME/CFS, that is readily transferable to diagnostic laboratories for routine tests on community-wide patient samples, is still urgently needed.

At present, a formal diagnosis is given only after eliminating all other diseases with similar symptoms, and with the range of self-reported symptoms fitting within defined sets of clinical criteria for a six month duration [28]. The difficulty for both patients and health practitioners has been exacerbated by over 20 different case definitions or diagnostic criteria for ME/CFS having been published to date [44]. With the underlying pathophysiology of ME/CFS still being unraveled today, there has been no gold standard against which to assess the effectiveness of each case definition. The 1994 Fukuda diagnostic criteria [45] developed by the Center for Disease Control in the USA, has been most commonly used by researchers and clinicians [46], yet it does not include the core defining symptoms of post-exertional malaise and neurocognitive disturbances, nor does it exclude patients whose symptoms may originate from a psychiatric disorder. The Canadian Consensus Criteria (CCC) [28] developed in 2003 by an international ME/CFS expert group was a significant improvement as it highlighted post-exertional malaise as a core symptom, along with fatigue, sleep dysfunction and pain. Additionally, neurological/cognitive and autonomic/neuroendocrine/immune symptom groups were included. In 2011, ‘International Consensus Criteria’ were formulated as a refinement of these CCC criteria, putting more emphasis on inflammation and neuropathology and focusing on neurological disturbance, immune/gastrointestinal, and energy [29]. A detailed review of the criteria used to diagnose ME/CFS was released in 2015 by the Institute of Medicine (IOM) of the Academy of Sciences USA and suggested a simplified core set of diagnostic criteria [47]. The IOM report acknowledged that the stigma associated with the symptom profile of ME/CFS had been due to a lack of knowledge and thereby also a lack of acknowledgement of the seriousness of the disease.

As there has been no single molecular biomarker test for ME/CFS, there are often long delays and high costs for the patient involved in the diagnostic process while diseases with overlapping symptoms are eliminated. There is also increased potential for misdiagnosis, so this context fundamentally impedes patient care. An important factor obstructing biomarker discovery has been the use of different diagnostic criteria from the many available to define the ME/CFS patient group under study, often preventing meaningful comparisons between research studies. Nevertheless, many potential diagnostic biomarkers have been identified by researchers, which indicate the involvement of improper immune function, inflammation and signs of autoimmunity. The requirement of specificity for ME/CFS and selectivity to prevent false positives or false negatives as with the development of all successful diagnostic tests is challenging. Most potential biomarkers identified so far are not easily transferable, however, to a simple universal diagnostic test.

The spectrum of subtypes of ME/CFS may hinder finding a universal test for all patients but targeting an early event in the development of the illness or a core feature of the illness may circumvent this. Despite these significant handicaps to diagnostic research, recent studies have emerged that have considerable promise for further development into a general accessible diagnostic test. Nanoneedle bioarray technology developed by Professor Ron Davis and colleagues measures a unique impedance signature that can distinguish moderate-to-severe ME/CFS from healthy controls. It measures electrical impedance modulations from cellular or molecular interaction in response to high salt concentration by utilising patient peripheral blood mononuclear cells or plasma. Using supervised machine learning algorithms, authors can identify new patients, an essential requirement for any robust diagnostic tool [48]. This is yet to be tested against other fatigue illnesses with overlapping symptoms to see whether it is specific for ME/CFS, but it also has the potential to identify a substance in the plasma of ME/CFS patients that is disease specific and itself would be a biomarker.

Another biomarker diagnosis based on three tests from Professor Paul Fisher’s group measures a combination of enhanced lymphocyte death rate as an initial screen, followed by more specialised assays of mitochondrial respiratory function; then, a cell-sensing kinase (Target of Rapamycin Complex1) in transformed lymphocytes (lymphoblasts). This combination gives high sensitivity and specificity for the accurate diagnosis of ME/CFS [49]. Despite its value and utility for selective confirmation of a diagnosis, the complexity of the specialist assays may mean it is not so easy to adapt or develop as a simple, universally available test in community diagnostic facilities.

We investigated the feasibility of measuring the chronic activation state of the stress RNA-activated protein kinase (PKR) for a diagnostic test. This kinase has been described as a ‘universal immunological abnormality’ in ME/CFS [50]. It could be used longitudinally over the first six-month period when a diagnosis is being made or as a biomarker to follow the course of the illness. PKR undergoes autophosphorylation when it is activated. In our pilot study, protein extracts of PBMCs from healthy subjects showed almost undetectable phosphorylated PKR, whereas extracts from ME/CFS patients had measurable phosphorylated PKR (pPKR) as analysed in a Western analysis. A ratio of the activated pPKR to inactive unphosphorylated PKR was determined in the patients who on average had had their ME/CFS for 10 years and the controls in the pilot study [51]. Although a very small cohort study, this revealed statistically (*p* < 0.05) or near to statistically significant differences between the two groups in lymphocytes and in neutrophils. Measuring the pPKR/PKR ratio has promise for development into an ELISA assay format, but it has yet to be shown whether it alone has the stringent specificity and selectivity that is needed for a diagnostic test.

The advent of so many Long COVID patients developing their post-viral syndrome at a similar time frame provides a useful study group to assess the feasibility of such an assay as an early diagnostic tool and for following the progress of the illness. It targets the biology of an early event in the development of ME/CFS that should be common to all patients, namely the shift from a transient immune/inflammatory response to a chronic response. It could be converted into a readily accessible simple test available to health practitioners in community diagnostic pathology laboratories. However, for real utility it needs validation with patients at the early stages of the illness (0–6 months) as well as of the specificity and selectivity with large patient numbers.

A test reported recently [52] that can be performed even more simply in a clinician’s office is an orthostatic stress challenge that shows different symptomatic, hemodynamic, and cognitive abnormalities in people with Long COVID and ME/CFS, compared with healthy controls. Coupled to a smartphone app to assess cognition, it can provide objective confirmation of the orthostatic intolerance and brain fog reported by patients with Long COVID and ME/CFS.

## 5. Holistic Model to Describe Both ME/CFS and Long COVID

To advance patient management further, there must be a greater understanding of the common defining features found in ME/CFS and Long COVID patients that (i) lead to a transition to a chronic state of an immune/inflammatory response, (ii) are the factors that prolong the long-term nature of the syndromes and facilitate the frequent relapses in the conditions and (iii) would enable an exploration of the unresolved detailed underlying mechanisms that lead to the array of symptoms. A confounding part of ME/CFS (and Long COVID) is that it is essentially a neurological condition and yet studying what is happening in the brain and CNS is incredibly difficult for most researchers and accessing cerebral spinal fluid from patients is quite invasive for those significantly debilitated.

We developed a holistic model for these illnesses to try to integrate the many interesting individual findings of the last decade or so from research studies, including from our own multi-omic molecular studies from our ME/CFS cohorts [13]. Most molecular studies have been restricted to using blood as the tissue source and their purified immune cells. Therefore, this raises the intriguing and challenging question as to how accurately results from peripheral tissues reflect molecular changes in the brain.

The first step in the development of ME/CFS (and Long COVID) occurs if the established transient immune inflammatory response of the peripheral system does not subside rapidly to a normal state, as happens in most people exposed to viral infections or transient life stresses. Then, the immune/inflammatory response becomes chronic and dysregulated with an ongoing perception of ‘a continuing threat’. What causes the activation of the immune system to become chronic? Certainly, the intensity and duration of the immune inflammatory response determines whether the immune signalling follows the typical course in healing or becomes destructive to normal physiology. While brief controlled inflammatory responses are beneficial in removing a threat such as a viral infection, if prolonged, as in ME/CFS and now suggested in Long COVID, prolonged activation of inflammatory cells and cytokines mean they continue to be dysregulated despite there no longer being any outside danger.

Autoimmunity and chronic viral infections are often found as a subtype/phenotype in ME/CFS. The regulatory CD4+ T cells (Tregs) have been proposed to be good candidates to be involved with the viral triggers of ME/CFS. Using ‘simulations of the cross-regulation model’ for the dynamics of Tregs, it was illustrated that mild infections might lead to a chronically activated immune response under control of Tregs if the responding clone has a high autoimmune potential [53].

Trained immunity has recently emerged as a new concept whereby the first line of defence against pathogens, the innate immune system, can unexpectedly also provide a non-specific immune memory [54]. Trained immunity may be inadvertently contributing to the risk of developing ME/CFS by producing higher levels of proinflammatory cytokines, thereby prolonging a chronic inflammatory state.

Trained immunity, coupled with hyperactivation of the innate immune system [53], could promote a secondary response to sustain Long COVID and ME/CFS as ongoing illnesses with a low-level chronic inflammatory state. Indeed, our transcriptome [55], proteome [56] and DNA methylome [57] studies of cohorts of ME/CFS patients who have had their illness on average for 10 years found consistent evidence of ongoing chronic activation/dysfunction of the immune /inflammatory system.

## 6. Evidence That Neuroinflammation Is Associated with ME/CFS

The next major question is how the brain and the CNS become involved with the development of the neurological symptoms that define the diseases and eventually become the dominant features and core symptoms. The chronic state of the peripheral immune/inflammatory system apparently results in atypical signaling to the brain and central nervous system that chronically activates and sustains the specific components of the CNS’s microglial mediated immunological/inflammatory response. There is initial evidence that this results in chronic neuroinflammation in both ME/CFS [58] and Long COVID [59], although more studies are needed to consolidate these findings. A key but isolated ME/CFS study gave evidence of neuroinflammation in the CNS using the non-invasive imaging technique, Positron Emission Tomography (PET) coupled with Magnetic Resonance Spectroscopy (MRS) together with a radioactive ligand for a translocator protein expressed in activated glial cells as a marker of neuroinflammation [58]. ME/CFS patients with varying severity of symptoms were compared with healthy controls to identify the brain regions where neuroinflammation was occurring. This approach is limited by the resolution of the technology at present. Neuroinflammation was occurring largely in the limbic system (cingulate cortex, hippocampus, amygdala, and thalamus), a region between the brainstem and the upper regions of the brain. The nearby midbrain and pons region of the brainstem were also potentially affected. Increased binding of the radioactive tracer was ~50–200% higher than in the healthy controls, and an important part of the outcome was a correlation between the severity of the ME/CFS symptoms and the extent of activation of the microglia. Chronically activated microglia promote inflammatory functions that lead to neurological dysfunction [60], characteristic of that seen in ME/CFS. From these observations we proposed neuroinflammation in the brain is fundamental for both sustaining ME/CFS and for facilitating the frequent more severe relapses of the illness in response to environmental, physical, emotional or psychological stresses [13,30]. A caution is that the ligand used in the ME/CFS study does not have absolute specificity, but more specific ligands have now been developed [61] and we await confirmatory studies.

Magnetic Resonance Spectroscopy (MRS) has been used to measure brain metabolites (choline, myoinositol, lactate, and N-acetyl aspartate) linked to inflammation and has determined that brain temperature is elevated in ME/CFS patients compared with matched controls. Increased metabolic ratios were found in the ME/CFS patients in 7 out of 47 brain regions compared with controls that correlated with fatigue, and increased temperature was observed in several brain regions. It was concluded the findings may indicate neuroinflammation [62].

The brain stem is a probable key player in neuroinflammation’s causes and effects. Recently it has been noted many ME/CFS symptoms can be linked to vital autonomic functions, for example, the brain stem plays a role in the regulation of heart rate, blood pressure and respiration. There is reduced white matter volume in ME/CFS in the pons, and the correlation pattern of the autonomic centre nuclei reflect autonomic centre dysfunction, such as postural orthostatic tachycardia syndrome, orthostatic intolerance, arrhythmia and abnormal breathing [63].

Our model proposes that abnormal signaling or transport of molecules/cells occurs through neurovascular pathways, and sensory neurons (in Figure 3 shown as inflammatory reflex and gateway reflex) and/or a leaky dysfunctional blood–brain barrier (BBB) to initiate the brain’s unique immune inflammatory response. It implies initial systemic inflammation can lead to, for example, inflammatory signals or even immune cells/molecules migrating into the brain. If it persists, strong signals are sent along peripheral afferent neurons, culminating in the neuroinflammatory response. The result is a compromised ability of the brain to regulate body physiology and disturbed homeostasis leads to the neurological symptoms of fatigue, brain fog, unrefreshing sleep and post exertional malaise characteristic of ME/CFS. We propose the central nervous system (CNS) through the hypothalamus/paraventricular nucleus and the brain stem can then signal back to the peripheral system to modulate much of the body’s homeostatic state and physiology through well-established pathways [13]. The resulting symptoms and the neurologically driven “sickness response” for the ME/CFS patient would persist, preventing a return to the pre-infectious/stress-related state. To maintain our integrity, information about aberrant bodily states is conveyed by interoceptive pathways [64]. Signals stimulated by hormones or cytokines from the body integrate with signals from the brain, assessing risk based on prior experience and resulting in adaptive sickness behaviours such as fatigue, promoting rest to conserve resources and social isolation to prevent infection transmission [64]. Chronic fatigue such as that experienced with ME/CFS patients is postulated to be due in part to comprised interoceptive processing accuracy as a result of an exposure to the environmental or social stressor. A constant cycling of molecular “danger signalling” between the systemic innate immune system and brain’s innate immune system may then be set up and persist to prolong the illness state in the long-term. In the brain, it has been proposed that danger signalling might occur through damaged mitochondria, acting as a signalling organelle [65], for example, with leakage of the energy molecule, adenosine triphosphate (ATP) and subsequent adenosine signalling to activate microglia.

One puzzling aspect of ME/CFS is what facilitates the sensitivity towards even minor day-to-day stresses that lead to relapses. Each is apparently interpreted by the ME/CFS patient’s physiology as a major danger that enhances neuroinflammation and results in a worsening of symptoms. The stress centre of the brain is in the hypothalamus within a cluster of neurons called the paraventricular nucleus. The PET/magnetic resonance imaging studies have not been sensitive enough yet to be able to show whether the hypothalamus is affected by neuroinflammation, but it seems likely. It is hypothesised that a neuroinflamed PVN may produce abnormally high amounts of the stress hormone, i.e., corticotrophin-releasing hormone. This triggers a series of events that can lead to deleterious excess serotonin production with complex effects on the brain. Excess serotonin is known to cause many of the neurological symptoms seen in ME/CFS [66]. Each incoming stress signal that is processed in the PVN may therefore act as fuel to keep the cycle of illness going by further activating the microglial immune/inflammatory response and dysregulation of brain/CNS function and body physiology.

## 7. Underlying Mechanisms Supporting a Disturbed Homeostasis

The inflammatory reflex:

The nervous system is composed of afferent (sensory) and efferent (motor) neurons. The afferent/sensory neurons relay signals from organ systems to the brain where the information is then used to send a response via efferent neurons to alter the organ systems’ functions depending on the body’s needs [67]. The autonomic nervous system controls the body’s involuntary systems via trillions of afferent nerves responsible for detecting slight changes in temperature, pressure, blood flow and small metabolic changes. This information, when received by the brain, elicits an efferent response to effect the appropriate change.

The sensory neurons detect minute molecular changes within the periphery and detect molecular changes indicative of systemic inflammation. This recognition of inflammation in the peripheries has been shown to stimulate the HPA axis and create a neuroinflammatory response in the brain, as systemic levels of inflammatory mediators can act directly on receptors in the brain, suggesting a link between systemic inflammation and neuroinflammation [67,68,69].

In response to infection, mast cells, dendritic cells and macrophages associated with paraganglia have been proposed to release IL-1 that acts on the brain, stimulating a neuroinflammatory response [70]. Neurons in the CNS produce and release cytokines responsible for modulating inflammatory responses to peripheral nervous system (PNS) neurons and vice-versa in a two-way flow, suggesting a pathway between PNS and CNS [67]. Microglia respond to peripheral inflammation by releasing inflammatory cytokines that can lead to a pathological state [71]. Although there is little understanding of how peripheral inflammation leads to microglial activation, it is possible that the inflammatory reflex plays a key role in this and therefore in the development of neuroinflammation in response to systemic inflammation, which could explain many of the symptoms seen in ME/CFS.

The Hypothalamus–Pituitary–Adrenal (HPA) axis:

Minor stressors such as a temperature change, a glass of wine or a walk around the block often trigger relapses in ME/CFS patients, suggesting that the biological mechanisms responsible for regulating this response to stress are no longer functioning normally. The hypothalamus–pituitary–adrenal (HPA) axis, a neurobiological stress system, is one of the most important mediators of the body’s response to stress. The HPA axis consists of the paraventricular nucleus (PVN) of the hypothalamus, the pituitary gland and the adrenal glands. In response to an initial stressor or chemical mediator such as serotonin, the PVN will release corticotropin-releasing hormone (CRH) and arginine vasopressin (AVP), which act on the anterior lobe of the pituitary gland, prompting the production of adrenocorticotropic hormone (ACTH). ACTH travels in the systemic circulation where it acts on the adrenal glands, leading to the release of cortisol [72]. Cortisol is important in regulating metabolism, mobilising glucose to the brain, modulating gluconeogenesis and glycogen synthesis in the liver and inducing protein degradation in muscle to facilitate gluconeogenesis. Cortisol also attenuates immune responses by suppressing B and T cell activity [73].

The HPA axis is responsible for mediating our inflammatory and immune responses and preventing them from damaging our own body. Hypoactivity/dysfunction of the HPA axis would mean ME/CFS patients’ protective response against the body’s inflammatory/immune reactions would be diminished [74]. In instances where the initial stressor is not alleviated, the HPA axis would be repeatedly activated, leading to cortisol surges that over time lead to a decrease in cortisol levels. CRH dysfunction, depletion of cortisol, glucocorticoid receptor resistance or impaired cortisol secretion have all been cited as possible causes for this drop off [75]. Repetitive bursts of inflammatory cytokines over a long period of time have also been shown to lessen the response of the HPA axis, showing that chronic systemic or neuroinflammation may lead to HPA axis dysfunction and hypocortisolism [76]. It has been suggested that these low levels of cortisol are the result of upregulated glucocorticoid receptors (GR) and mineralocorticoid receptors (MR) that control cortisol release through negative feedback loops by inhibiting CRH and ACTH release [75]. If these receptors are upregulated and the HPA axis is chronically activated over time, the negative feedback on the HPA axis may lead to downregulation of the system and a reduction in sensitivity to stress [72]. This HPA axis desensitisation is known as adrenal fatigue or cortisol dysfunction. Patients with this condition experience symptoms such as myalgia, fatigue, memory loss, brain fog and orthostatic hypotension [75]. These symptoms may contribute to those suffered by ME/CFS and Long COVID patients.

Serotonin regulation:

The body’s serotonin release in response to stress is regulated by two CRH receptors (CRHR1 and 2). In periods of high stress, CRH is released from neurons in the PVN in large quantities. This leads to the internalisation of CRHR1 from the membranes of serotonin-releasing neurons and replacement by CRHR2 [77]. CRH acts then predominantly on CRHR2, leading to raised levels of serotonin. Therefore, only at high levels of stress is excess serotonin found in the body, assuming all serotonin regulatory systems are functioning. Serotonin in excess leads to many abnormal symptoms that are indicative of a loss of control over its normal functioning. These symptoms include dysfunctional muscle contraction through motor neuron inhibition, migraines, sleep disruption, dyspnea, hyperalgesia and cognitive dysfunction [77]. Serotonin can also lead to the release of dopamine and norepinephrine, which may explain other physiological changes experienced in ME/CFS linked to memory, gastrointestinal problems, mood and even blood coagulation. Furthermore, it has been shown that dopamine release and its self-oxidation to aminochrome can lead to mitochondrial dysfunction, which may relate to mitochondrial abnormalities reported in ME/CFS patients [78].

Serotonin itself can lead to the release of CRH from the PVN of the hypothalamus. Therefore, chronically high levels of serotonin could lead to chronic reactivation of the HPA axis [79]. This is another potential mechanism for HPA axis dysfunction. The combination of both HPA axis dysfunction and excess serotonin would explain many of the symptoms found in ME/CFS patients; fatigue, pain sensations, mood swings, muscle tightness, insomnia, and cognitive disruption (often called brain fog) and sensory hypersensitivity. Excess serotonin can affect the permeability of the blood–brain barrier (BBB), so this could enhance the microglia-fed neuroinflammatory response.

These are but two features of the disturbed physiology that we are beginning to understand in terms of their important roles in the pathophysiology of ME/CFS and Long COVID.

## 8. The Importance of Blood–Brain Barrier Permeability

The blood–brain barrier (BBB) is composed of tight junctions formed by endothelial cells that function to prevent substances, cells or molecules from diffusing into the neural tissue of the CNS and causing irreparable damage. Blood–brain barrier permeability (BBBP) is controlled by organised functional units made up from microglia, pericytes, astrocytes and the basement membrane [71]. It has been proposed that increases in BBBP in patients with ME/CFS could lead to many of the symptoms experienced by these patients [66]. If any factor were to lead to dysfunction in the microglia, astrocyte, pericyte and basement membrane functional units this would have a significant impact on the BBBP. Immune cells and neurotoxic molecules would be allowed to enter the brain, leading to an immune response in the brain and neuroinflammation, precipitating the symptoms of ME/CFS.

To date, many mechanisms have been explored that show increases in BBBP can and do lead to disease, for example as a secondary response to stress via excess serotonin release [66]. Systemic inflammation can increase BBBP with microglia migrating to cerebral vessels [71]. Astrocytes can be directly responsible by secreting IL-17A, resulting in downregulation of claudin-5 produced in microglia, a molecule playing a key role in preventing BBB permeability. If a dormant virus such as the Epstein Barr virus were to be reactivated in BBB endothelial cells, this would lead to increased BBBP, providing a possible mechanism for how viral reactivation could have a role in ME/CFS [80].

## 9. Can Viral Reactivation Affect ME/CFS?

Infection with the Epstein Barr virus is not only a common initiator of ME/CFS, but also exists in a latent state in most of the population. There is increasing evidence that reactivation of the virus can play a role in prolonging both ME/CFS and Long COVID. Accumulating data suggest that EBV can exist in an intermediate abortive lytic/leaky replication state [81], and that expression of a specific EBV-encoded protein, dUTPase, can influence the expression of many genes associated with blood–brain barrier (BBB) integrity [82], can modulate synaptic plasticity and thereby also affect cognitive processes and promote pro inflammatory cytokines known to disrupt the BBB. This could be significant in a phenotype/subtype in ME/CFS to promote and maintain neuroinflammation [82]. For example, dUTPase can induce the increased secretion of multiple cytokines and chemokines including TNF-α, TGF-α, IL-1β, IL-6, IL-8, IL-12p40, IL-23, CCL5, CCL20 and IFN-γ in PBMCs [82]. It can increase expression of COX-2, a molecule that is implicated in neuroinflammatory toxicity in astrocytes and microglia [83] and modulate tryptophan and thereby serotonin metabolism that can affect the BBBP. Activation of the Epstein Barr virus, for example, might be destructive in the pathophysiology of ME/CFS and its frequent relapses.

Other viruses have been implicated as viral triggers of ME/CFS. Prusty and colleagues [65], using an epithelial (U2OS) cell culture Human Herpes Virus-6 (HHV6) latency model recently identified an early stage of HHV-6 reactivation, termed here as transactivation, that is characterised by the transcription of several viral small non-coding RNAs but with the absence of increased viral replication. These data suggest a lytic leaky/replication state might be occurring in HHV-6 as well. Following transactivation, in cells carrying latent HHV6 virus, mitochondria were shown to be fragmented and, dUTPase was strongly induced. The transfer of serum from patients with ME/CFS in the cell culture model produced a similar fragmentation of mitochondria. HHV-6 reactivation in ME/CFS patients activates a multisystem, proinflammatory, cell danger response but comes at the cost of mitochondrial fragmentation and severely compromised energy metabolism [65]. HHV-6 transcripts were analysed in postmortem tissue biopsies from a small cohort of ME/CFS patients and matched controls by fluorescence in situ hybridisation and showed abundant viral miRNA in various regions of the human brain and associated neuronal tissues including the spinal cord in ME/CFS patients but not in controls [84].

Viral reactivation may then be a contributor to ongoing ME/CFS disease and frequent relapses in some patients and therefore may form a definite subgroup that could be targeted with antivirals if they are able to suppress the leaky/lytic replication state.

## 10. Can the Microbiome State Be Manipulated to Improve ME/CFS Pathology?

In the search for ways of improving outcomes for ME/CFS and Long COVID patients, manipulating the microbiome may be an important approach.

The microbiome has many important roles integral to maintaining human health. The microbiota is responsible for the production of vitamins, absorption of ions, regulation of metabolism, development of gastrointestinal mucosa, production of antimicrobial substances and the health of the brain. Dysbiosis/loss of variation of microbiota has been linked to disease and may play a role in ME/CFS [85,86,87]. The gut microbiome has been shown to be important in many parts of human health, such as for the brain, the control of blood glucose and the prevention of obesity. Microbial diversity is lower in autoimmune conditions [88]. Various theories and ideas have been proposed for the involvement of the gut microbiome in ME/CFS pathophysiology: (i) an alteration in intestinal microbiota and resulting dysbiosis, (ii) alteration in gut–brain communication, (iii) a leaky gut, (iv) D-lactic acid-producing bacteria inducing neurological symptoms, (v) an alteration in kynurenine production from tryptophan and (vi) past antibiotic use. The multiple theories are supported by a plethora of studies of often conflicting data suggesting the need for comprehensive carefully controlled studies to define which of the theories might be important in ME/CFS.

1. Alteration in intestinal microbiota and resulting dysbiosis: There is clear evidence of changes in microbiota composition and dysbiosis, but findings are inconsistent and the role in the pathophysiology of ME/CFS still remains unclear, although an increase in blood inflammatory markers has been documented [89,90]. Probiotic use by ME/CFS patients suggested improvement in cognitive function [91,92,93] and rectal infusions of bacteria [94] also improved ME/CFS symptoms. Studies that have been published only in *Cell Host & Microbe* are very informative. The reduced abundance of health-promoting butyrate producing bacterial species is linked to symptom severity, providing a better understanding of the negative consequences of dysbiosis [95]. Oh’s research group identified phenotypic, microbial and metabolic markers specific to patient cohorts. ME/CFS of shorter duration showed the distinct depletion of butyrate-producing bacteria dysbiosis, whereas in the longer-term patients this was not so evident, but they showed severe metabolic abnormalities [96].

2. Leaky gut: A leaky gut caused by epithelial barrier dysfunction is thought to be induced by inflammation and in ME/CFS patients may lead to irritable bowel (IB) symptoms, or IB conversely may predispose the development ME/CFS. A leaky gut with increased permeability may support the hypothesis that there is bacterial translocation in systemic circulation [97,98,99,100]. 

3. D-lactic acid: Excessive D-lactic acid accumulation from bacterial fermentation increases the concentration of the molecule in the blood and the brain, which is hypothesised to cause neurological symptoms [101,102].

More studies are needed to clearly define these mechanisms. Therapeutic interventions with probiotics or fecal transplant in general have not revealed consistent results, although one fecal transplant study showed a significant improvement [103]. There is broad consensus of a “need for better study designs including a consistent use of the case definition in research, a higher study quality as well as more longitudinal studies” [104]. The microbiome may prove to be an important component in the pathophysiology of ME/CFS and Long COVID.

## 11. Can the Focus on Long COVID Research Provide Benefit for ME/CFS Patients?

The emergence of Long COVID cases in such large numbers from the current pandemic has resulted in an investment into research of a size that has never been seen before for a post-viral disorder such as ME/CFS. It has been accompanied by an intense broad-based focus and interest from scientists and clinicians that ME/CFS patients never experienced. We do not know yet whether Long COVID will prove to be as prolonged as ME/CFS for some or most of those affected. Even if the research investment on Long COVID focuses on the SARS-CoV-2 cohort exclusively, as is the current trend, the current research activity should also be of huge benefit for ME/CFS patients. Already documented is the close similarity of this specific post-viral syndrome with ME/CFS, and that means any advances in reversing or even ameliorating the neurological symptoms with Long COVID will likely transfer over to at least some of the phenotypes of the ME/CFS group. The fact that Long COVID is in its relative infancy and with large numbers means it lends itself to what has been lacking in ME/CFS longitudinal studies.

Longitudinal studies:

Most of the studies to date on ME/CFS have been single-time-point patient studies only capturing a small time-window of an illness that is lifelong. However, diagnosis takes place only after six months of symptoms, and new patients drip feed into the community with endemic viruses such as Epstein Barr or individual stressful events in the lives of those affected. There has never been a large cohort of new ME/CFS patients available in such a short timeframe as there is now with Long COVID patients. This means long-term longitudinal studies that follow the course of ME/CFS through the patient’s lifetime are not easy to establish, and they are lacking. Long COVID patients form a natural large group that could be studied in this way. Even the ME/CFS and Long COVID syndromes have varying phenotypes/subtypes, they may not, however, suit a universal course of action for patients, particularly if there are variable frequencies of particular phenotypes in different cohorts. Sadly, as with many diseases, those with a rare form/phenotype of ME/CFS may be overlooked by studies and therefore miss out on the full benefits of future therapies and studies. Minority phenotypes not identified within studies may have varying results and change conclusions of studies undertaken. Key features for the benefit of patients may be smoothed out and missed if they are in minority phenotypes, and conclusions may therefore vary according to the composition of the cohort.

Therapeutic possibilities:

Currently, there is a dearth of therapeutic options for ME/CFS patients and promising treatments have rarely resulted in validating trials that enable more comprehensive studies. When a promising treatment does not benefit most patients, impetus for further studies is often lost. Since combinations of treatments might be beneficial, trials of a promising candidate therapeutic in combination with other compounds are needed. There are examples of small studies with such combinations that are indeed showing promise, including CoQ10 and Se [105] and CoQ10 and lipoic acid [106].

Our model for ME/CFS and Long COVID [13] highlights areas where therapeutic intervention might be helpful: (i) the switch from a transient to chronic inflammation response; (ii) chronic fluctuating neuroinflammation; (iii) the management of the stress response; (iv) overproduction of serotonin; (v) reactivation of viruses; (vi) manipulation of the microbiome (once the connection with the development and/or maintenance of the chronic post-viral state is better understood) and (vii) overcoming energy production dysfunction. Figure 4 summarises the key features of the post viral/stressor syndromes, ME/CFS and Long COVID and highlights areas where therapeutic options as described below can be exploited.

The key features of the syndromes are shown in red, including involvement of subtypes/phenotypes that might be exploited for tailored therapeutic options. Key areas of physiology that are targets for therapies are shown in blue, with some examples of therapeutic possibilities. Many aspects of the dysfunctional physiology such as chronic HPA axis dysfunction, oxidative stress and serotonin regulation are possible key features of the development of the disease but also features that are dysfunctional to sustain ME/CFS and Long COVID as ongoing syndromes. There may be a connection between the microbiome and brain dysfunction but that is still to be resolved.

1. Immune dysfunction: There has been a wide application of strategies to combat immune-mediated inflammatory diseases generally over the last 40 years, starting with broad-spectrum immune modulators and evolving to highly specific drugs to manage chronic immune/inflammatory diseases with increasing specificity from highly targeted medicinal chemistry. This development was discussed elegantly along with future prospects in McInnes and Gravallese (2021) [107]. The approaches have been effective in some diseases but not in others [107]. For ME/CFS, Rituximab, which depletes B cells, looked to be a promising therapeutic for ME/CFS in early trials, but unfortunately was not associated with clinical improvement in a phase 3 trial [108,109,110]. There are many immune targets that are being investigated in other diseases and many are in clinical trials. ME/CFS and Long COVID patients may eventually obtain indirect benefits from these trials.

2. Neuroinflammation: Low-Dose Naltrexone (LDN) appears to be able to regulate the activity of microglia immune cells in the central nervous system and control the pro-inflammatory factors causing neuroinflammation, and it has been shown to be a promising drug to modulate symptoms in ME/CFS [111]. A retrospective study utilising the medical records of 218 patients suggested about 75% reported a positive response in some of their neurological symptoms [112]. LDN has also been found to restore functional activity of Transient Receptor Potential Melastatin 3 (TRPM3) in Ca^2+^-dependent Natural Killer (NK) cells of ME/CFS patients [113]. A safety and efficacy trial of LDN in a Long COVID cohort of 53 patients concluded LDN was a safe drug for this patient group and may reduce symptoms of Long COVID. LDN remains an important candidate to target neuroinflammation.

3. Stress: Managing stress requires a functional HPA axis and avoidance of high or low cortisol levels, all of which are not the typical states in ME/CFS. Critical is the corticotrophic-releasing hormone (CRH) released from the paraventricular nucleus (PVN) of the hypothalamus, whose activity is modulated with glucocorticoids [80]. The complex set of interactions mediated by the HPA axis seem incredibly difficult to target therapeutically. However, if the postulated chronic fluctuating neuroinflammation can be controlled, the PVN may not produce the excessive amounts of CRH in response to even minor stresses, as seems likely in ME/CFS, and that accounts for patients’ sensitivity to stress. Hence, targeting neuroinflammation might be sufficient.

4. Excessive serotonin: Serotonin is also a regulator of the HPA axis through the stimulation of CRH. It has been proposed that excess serotonin could in whole or in part be an explanation for the symptoms of ME/CFS [114]. The essential amino acid tryptophan is the precursor to serotonin via 5-OH tryptophan. Tryptophan has a two-enzyme system (IDO1 and IDO2) that acts to control its availability for serotonin synthesis through the kynurenine pathway responsible for converting tryptophan into nicotinamide adenine dinucleotide (NAD^+^) [114]. Both IDO1 and IDO2 are present at the same point in the pathway and catalyze the same reaction but their activities are dependent on the concentration of tryptophan. IDO2 becomes active when tryptophan is excessive. The gene for IDO2 has common mutations within the population, some of which are inactivating, so some people will lack this fail-safe system. For an ME/CFS patient, it would put them into a metabolic state with the potential for excessive levels of tryptophan and thereby excessive serotonin. This has been proposed as a disease model for ME/CFS patients [115]. We have shown both healthy controls and ME/CFS patients have the same frequency of the inactivating mutations, so it may be a silent mutation that has deleterious effects only for susceptible people who have developed ME/CFS or Long COVID [1]. Therapeutic targeting of the kynurenine pathway if an ME/CFS patient has an inactivating mutation in IDO2 might be beneficial.

5. Reactivated viruses: Given the resurgence in interest in the importance of reactivated viruses, such as Epstein Barr and HHV6, antivirals that have been prominent in amelioration of the serious effects from COVID-19 might have a place in the therapeutic tool bag for a phenotype/subtype(s) of ME/CFS and for Long COVID.

6. The microbiome: Establishing the important mechanisms of the connection between the changing microbiome and ME/CFS and Long COVID and whether it is a cause or consequence of the syndrome is important. Is there an initial altered microbiome in susceptible people that facilitates the chronic immune response and development of ME/CFS, or does the onset of the chronic condition alter the microbiome that then might be part of sustaining the ongoing disease? With this knowledge there may be important therapeutic options that could be pursued.

7. Energy Production: Defects in energy production in ME/CFS have been well established [1] and supplements to boost its health are in common use. However, there have been no clinical trials with ME/CFS to document benefits of one or a combination of compounds. Within the phenotype/subtype model there is a real need for trials, as energy insufficiency, oxidative stress and reactive oxygen species are all features of ME/CFS and Long COVID syndromes. Paul et al. [116] have speculated that the symptoms of both Long COVID and ME/CFS may stem from redox imbalance, which in turn is linked to inflammation and energy metabolic defects. Treatments to restore redox imbalance in the body may involve stimulating endogenous defence mechanisms or mimicking them to restore balance where it is needed. As mitochondrial dysfunction is responsible for producing increased ROS and contributes to the symptoms of ME/CFS, this organelle would be a key target for treatment. Studies on ME/CFS patients have suggested some may have a deficiency of Coenzyme Q10 (CoQ10), a state known to lead to a reduction in energy production and also increased production of ROS [105,117]. Idebenone (a CoQ10 analogue with better absorption and bioavailability) improved chronic fatigue in multiple sclerosis patients [118]. MitoQ, an analogue of CoQ10 specifically targeted to mitochondria, also has promise as an antioxidant therapy for these conditions [117]. Cytoflavin (a complement of inosine, nicotinamide, riboflavin, and succinic acid) was used to target patients undergoing rehabilitation following SARS-CoV-2 infection and aimed to reverse mitochondrial dysfunction. In the study it reduced weakness, fatigue and breathlessness [119]. Antioxidant drugs have been approved for use in other neurological diseases; for example, dimethyl fumarate (DMF) is a nuclear factor erythroid 2-related factor 2 (NRF2) activator with immunomodulatory and antioxidant properties commonly used to reduce fatigue in relapsing multiple sclerosis [120]. There are currently many other antioxidant drugs in clinical trials for other conditions, including sulforaphane (an NRF2 activator), ebselen (a glutathione peroxidase analogue) and GC4419 (a superoxide dismutase analogue) [121]. Natural and synthetic supplements also have been claimed to have benefits in improving redox imbalance, including curcumin (turmeric extract), resveratrol and broccoli extract, with trials ongoing [122,123,124]. Vitamin E and C have both been shown to reduce oxidative stress and can work synergistically [125,126].

Various antioxidants, then, have promise as therapeutics for ME/CFS and Long COVID to redress redox imbalance and for ameliorating distressing symptoms patients experience with these post-viral conditions. CoQ10 and its analogues (MitoQ and idebenone) as well as approved neurological drugs, dimethyl fumarate and edaravone, can control excessive ROS. Precursors to the body’s most abundant antioxidant glutathione, such as N acetylcysteine, can redress a deficit in the natural antioxidant that has been deduced to be deficient in brain imaging studies. These antioxidants should be evaluated in clinical trials with Long COVID and ME/CFS patients to determine if there are therapeutic benefits.

Clinical trials and new therapeutics

The large investment in Long COVID research through the National Institute of Health’s Recover project brings with it the hope of much-needed clinical trials to expand the therapeutic tool kit available for those suffering with these post-viral and stressor syndromes. The modest investment in ME/CFS research and lack of broad interest in the past has severely limited these opportunities but there should be a flow of benefits from the Long COVID trials even if they do not include ME/CFS patients. Additionally, the renewed interest that Long COVID has stimulated across the multidisciplinary scientific community brings new opportunities as well as the chance to explore new possibilities. In New Zealand, a plant extract has been discovered that could reverse neuroinflammation in a zebra fish and a mouse model of obesity and glucoregulation, and also showed positive benefits in a first human trial. This stimulated us to plan first-stage testing with ME/CFS and Long COVID patients to determine whether the extract can improve the neurological symptoms and show any molecular changes in the signatures of the syndromes [127].

## 12. Patient Lifestyle Self-Management of ME/CFS and Long COVID

Most ME/CFS patients learn by experience how to manage their condition as best they can, given the lack of therapeutic options and incomplete knowledge of the illness of their clinicians. This can involve diet, supplements, relaxing modalities such as yoga and meditation and other strategies that provide them incremental improvements. In our experience and through contact with patients, we have seen the ME/CFS community show impressive resilience and a desire to improve their own lives using novel coping strategies, despite the currently bleak outlook in having this illness. For many years, graded exercise and cognitive behavioural therapy were recommended as the preferred treatment options despite post exertional malaise being a core symptom, with reinforcement from the PACE trial in the United Kingdom [128,129]. The results of the PACE trial have been discredited by biostatisticians, and finally in 2021 the Nice guidelines [130] removed these treatment modalities as the preferred ones, and this was simultaneously re-enforced by specialist ME/CFS clinicians in North America as well.

Intense cognitive behavioural therapies such as the Lightning process [131] have continued with promises of a high cure rate but without independent validating evidence. The claims seem incompatible with a phenotype/subtype model for ME/CFS discussed here. Whereas some patients obtain benefit, others are often left with incredible guilt for failing to cure themselves, and so these therapies can be counterproductive. Evidence-based trials are lacking. However, behavioural therapies have an important place, if transparent, and are an affordable option that might provide possible benefit to patients incrementally, and not as an instant cure. One such treatment option was brought to our attention recently by a New Zealand patient who, from a low place, obtained incremental benefit over several months from a programme, ‘dynamic neural retraining system’, based around recognising the brain’s neuroplasticity and potential to change its circuitry. There exists the potential to reverse patterns of interoceptive pathways that maintain a ‘sickness response’ where the brain circuity is wired to perceive ‘ongoing danger’ as a result of hormonal or cytokine signalling [64]. The ‘Dynamic Neural Retraining system’ was developed by Annie Hopper, and described in her book, ‘Wired for Healing’ [132]. She managed to reverse her severe neurological symptoms. Now this programme is offered publicly so others with chronic conditions may possibly benefit. The ME/CFS blog site Health Rising recently highlighted one case of a woman who benefited from this programme. Though initially wheelchair-bound with her ME/CFS, she returned to running again in a year after embracing the programme [133].

With the novel intense interest and research effort to understand Long COVID, it is hoped that a more systematic focus on ME/CFS will evolve, be recognised and be included in health and social initiatives in communities and countries worldwide. Currently, most of the knowledge and experience of ME/CFS rests within the ME/CFS patient group itself. They have much to offer those suffering from Long COVID regarding how to manage and cope with the illness. Now, with final acceptance that ME/CFS and Long COVID illnesses do not sit in the psychological/psychiatric diagnostic box, it is time for focused, consistent medical education of all clinicians, evidence-based best-practice clinical management—as outlined in the Mayo Clinic publication from the top ME/CFS clinicians in the USA [134]—and empathetic health and social support programmes that include new therapeutic options available to all patients. No longer should ME/CFS patients be the ‘missing millions’.

## Figures and Tables

**Figure 1 ijms-24-05124-f001:**
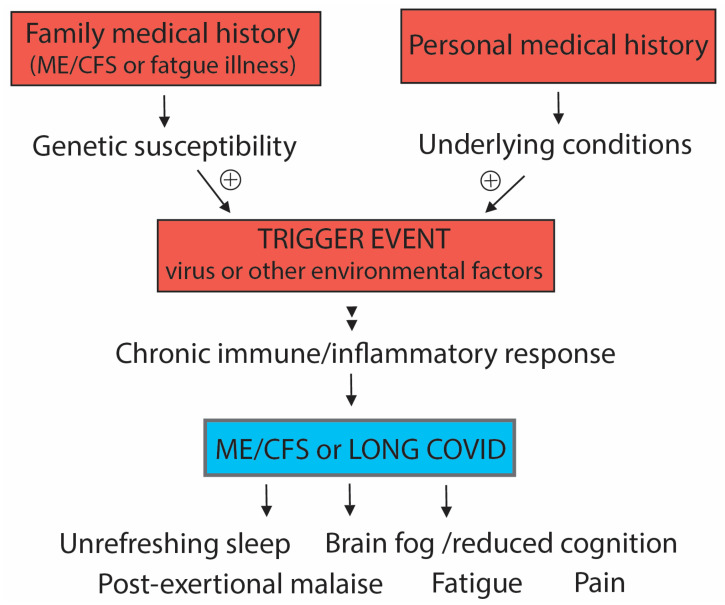
Features leading to the susceptibility of a person to develop a chronic response to a stressor and the lifelong illness of ME/CFS.

**Figure 2 ijms-24-05124-f002:**
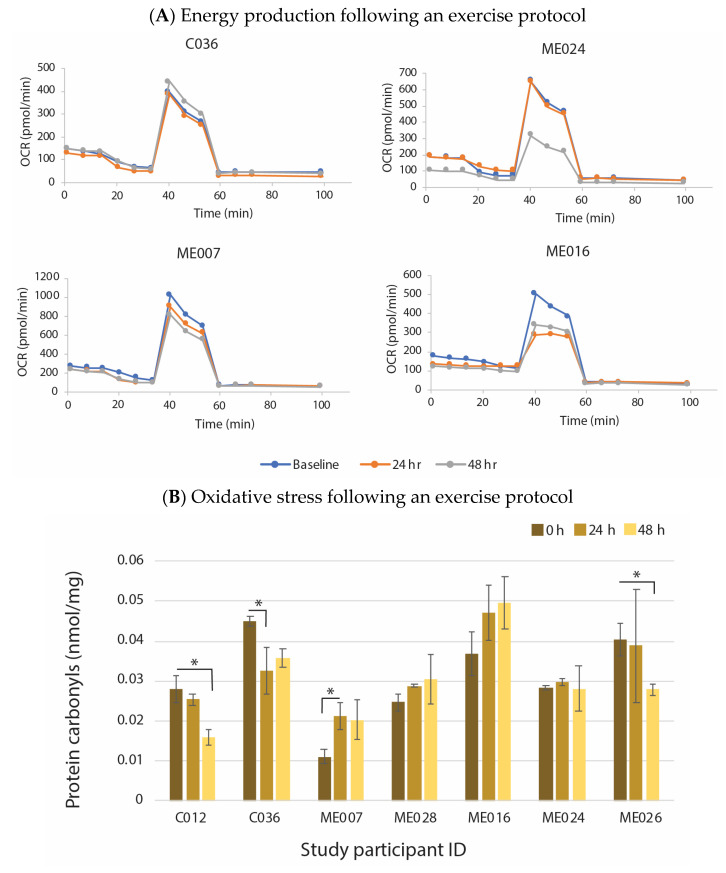
Molecular changes in individual ME/CFS patients after an exercise protocol, and in the DNA methylome of ME/CFS patients during a relapse/recovery cycle. (**A**) Energy production following an exercise protocol. Stress test profiles measured as changes in oxygen consumption on a Seahorse analyser of patient and control PBMCs isolated before exercise and 24 h after each of two exercise protocols (i.e., at 24 h and 48 h). Oligomycin, FCCP and Rotenone/Antimycin A were injected across the time course to stimulate or inhibit different actions of mitochondrial function and allow calculation of mitochondrial parameters [41]. Initially, basal respiration is measured (0–15 min), then ATP production deduced (15–35 min) and maximal respiration determined (40–60 min). Reserve capacity, protein leak and non-mitochondrial respiration can be calculated from the profiles. (**B**) Oxidative stress following an exercise protocol. Individual patient study of post-exertional malaise in ME/CFS patients (ME) and healthy controls; (**C**) assessed protein carbonyl modification of plasma proteins before and 24 h following each of two exercise sessions 24 h apart (24 h and 48 h). C012 and C036 are controls, and ME007, ME028, ME016, ME024, and ME026 are ME/CFS patients. All participants were young women in their 20’s. Error bars represent the SEM and a two-tailed students *t*-test determined significance of changes from before exercise in each case (* indicates a *p*-value ≤ 0.05). C012, C036 and ME026 showed significant reductions, ME007 a significant increase, while ME016 and ME028 trended towards significance increases. (**C**) DNA methylome changes during a relapse and then recovery. Methylation percentages are shown across five time points spanning a year of sampling (a to e) at three unique sites for each patient where methylation changed during relapse and then recovered. Patient 1 (top block) had a relapse over two time points (green shaded section), and patient 2 (lower block) had a relapse spanning only one time point (orange shaded section). Chromosome and co-ordinates for the site are shown above each section of the block. Percentage methylation is shown on the abscissa. (**D**) Methylome changes affecting functional changes during a relapse. Sankey plot showing the relationship between the variably methylated fragments (iVMFs) identified in each patient associated with the relapse event and the biological functions they associate with through various regulatory genomic elements of relevant genes. From the statistically significant variably methylated fragments identified for each patient, the locations were determined and relevant regulatory interactions recorded from the UCSC genome browser. A gene list was compiled of genes associated with these regulatory interactions and the functional annotations were utilised to place them into categories.

**Figure 3 ijms-24-05124-f003:**
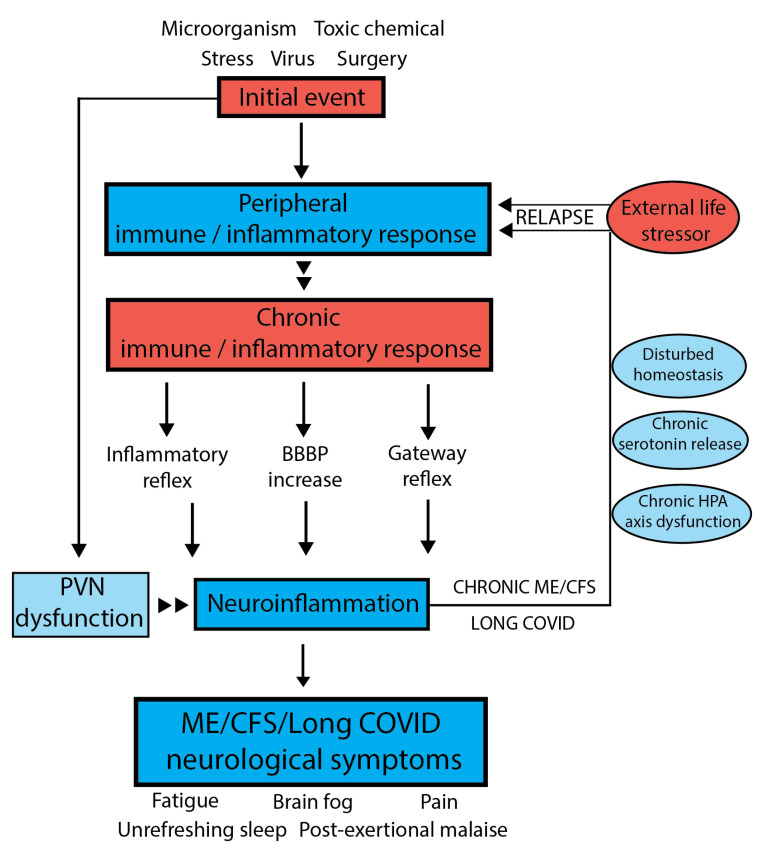
Model for the onset of ME/CFS and its progression to a chronic sustained illness with relapse/partial recovery phases. Following an initial external stressor event, systemic immune/inflammatory responses are activated, become chronic and these signals are communicated to the CNS via inflammatory and gateway reflexes and possibly an increase in permeability of the BBB. Neuroinflammation is activated affecting the stress center within the PVN of the hypothalamus and leads to a wide range of neurological symptoms that feedback to the periphery via disturbance of homeostasis and the stress activated HPA axis that becomes dysfunctional with chronic activation. The systemic physiology and molecular homeostasis are then chronically affected through important cellular functions such as mitochondrial energy production, metabolic activity and a continuation of immune/inflammatory reactions. External life stressors that feed into a disturbed PVN not only maintain the ME/CFS but also act to precipitate relapses. Modified from Tate et al., 2022 [13].

**Figure 4 ijms-24-05124-f004:**
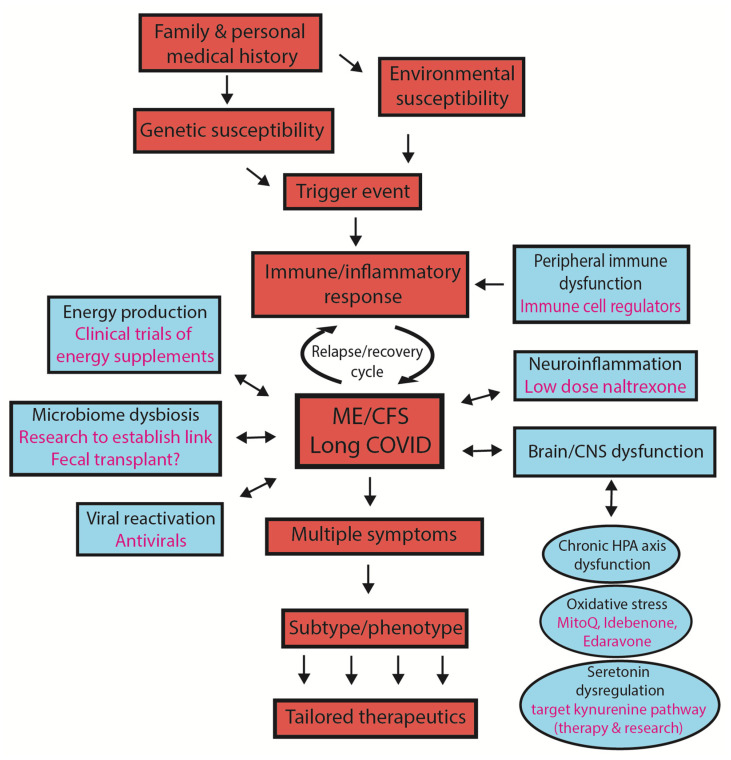
Therapeutic opportunities for ME/CFS and Long COVID management.

**Table 1 ijms-24-05124-t001:** (**A**) Severity of initial COVID-19 infection and initial triggers for participants’ ME/CFS. (**B**) Underlying health conditions.

(**A**)			
**Severity of COVID-19 Infection**	**Long COVID Participants, *n* (%)**	**ME/CFS Trigger**	**ME/CFS Participants, *n* (%)**
ICU/Hospitalised	5 (8.8)	Bacterial infection	4 (2.5)
Bed-bound	26 (45.6)	Viral infection	86 (53.8)
House-bound	16 (28.1)	Immune system problems	2 (1.3)
Mild symptoms	9 (15.8)	Emotional trauma	7 (4.4)
No Symptoms	1 (1.8)	Physical trauma	6 (3.8)
		Other	55 (34.4)
**Total**	57 (100)	**Total**	160 (100)
(**B**)			
**Underlying Health Conditions**	**Long COVID Participants, *n* (%)**	**ME/CFS Participants, *n* (%)**	**Total, *n* (%)**
Gastrointestinal issues	19 (33.3)	73 (45.6)	92 (42.4)
Mental health condition	10 (17.5)	58 (36.3)	68 (31.3)
Allergies	9 (15.8)	27 (16.9)	36 (16.6)
Autoimmune condition	8 (14.0)	25 (15.6)	33 (15.2)
Inflammatory disease	9 (15.8)	18 (11.3)	27 (12.4)
Lifelong low energy	2 (3.5)	24 (15.0)	26 (12.0)
Cancer	0 (0.0)	3 (1.9)	3 (1.4)
Other	30 (52.6)	69 (43.1)	99 (45.6)
None	11 (19.3)	25 (15.6)	36 (16.6)

## Data Availability

The datasets [40] used for analysis in Figure 2C are available in the GEO database NCBI (GSE166592).

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
