# Peer review of "Towards a Better Understanding of the Complexities of Myalgic Encephalomyelitis/Chronic Fatigue Syndrome and Long COVID"

_ijms, 2023, doi:10.3390/ijms24065124_

Round 1

Reviewer 1 Report

This review manuscript of Tate et al. is informative and important. I have only minor comments. 

1.       The number of ME /CFS patients in Table 1 of 160 is not consistent with the 161 patients described in the text (L61).

2.       The Reviewer would like to know what the authors think about ME/CFS that occurs after mRNA vaccination, which is often experienced clinically.

Author Response

Thank you for your encouraging review.

  1. Is the patient number 160 or 161? Thanks for noticing that discrepancy- I have checked the original source docs and it is 160 as in the Table. I have amended the text.
  2. ME/CFS that occurs after mRNA vaccination which is often experienced clinically

I am interested in this point and we are planning a study of those ME/CFS who had no serious reaction to the vaccine vs those who did, and if possible include if possible example of those who as a healthy person developed an ME/CFS like illness . I have included a section to ‘express my thoughts’.

“Interestingly, vaccination can be a trigger for developing ME/CFS in healthy people, for example, with the influenza vaccine. Now there are anecdotal reports from clinicians who have also seen the onset of ‘Long COVID’ or ME/CFS like illness following vaccination with the RNA vaccines (in New Zealand the BioNTech/Pfizer RNA vaccine has been almost exclusively used). Presumably, these affected people either carry a ‘silent’ genetic susceptibility that is challenged with the reactive vaccine, or their current state of personal health from previous environmental exposures to viruses, other microorganisms, or toxic chemicals makes them at least transiently particularly vulnerable to reacting in this way to the vaccine. For those people with ME/CFS already who showed a high susceptibility for a significant ongoing relapse after vaccination, it is assumed their chronically activated immune system is close to a critical threshold and the reactive vaccine is sufficient to drag the illness back into a more severe state. ”    

  1. Additionally the day after submission two excellent studies on the microbiome were published and I have added discussion of these studies with references .

Reviewer 2 Report

The paper by Tate et al. is a review article that discusses the complex nature of ME/CFS. The manuscript provides very useful information about ME/CFS and Long COVID and is generally well written. There are no major concerns with the manuscript; however, there are a few minor issues that need to be addressed.

1.     Any counting numbers less than 10 should be spelled out. For instance, in line 21, 6 months should be six months.  This is customary in scientific writing to spell out all counting numbers less than 10 unless they are attached with a specific unit such as 3 mL or 4 oC. Please check it throughout the manuscript.

2.     In a couple of places, viruses or other things are stated to be the “cause” of ME/CFS, where in fact, these are associations. I would strongly advise against using the word “cause” as there is not enough data to make such a strong claim at this time. Trigger is a bit better but still a bit strong.

3.     Some data from studies conducted by the authors are presented. There should be more discussion regarding the limitations of the studies or if available, any data from studies that do not support these conclusions.  

4.     Toward the end of the manuscript, the paper is written in the “first person” but given there are multiple authors on the manuscript, so either the person making the statement should be articulated or it should not be written in the first person (i.e. line 1008)

5.     I’m not sure why there is an Institutional Review Board Statement in a review paper given that it would have been given in the original manuscripts. Was original data presented in the review that was not articulated?

6. The graphics seem to be poor resolution. Please make sure they meet the requirements of the journal.

7.     Lastly, there are number of typos and formatting problems with the manuscript that should be fixed before its accepted. Also, hyphens are used when not necessary. For instance "microorganism" is not hyphenated nor is "preexisting" or "neuroinflammatory". Please check to make sure that other instances are addressed.

Thank you!

Author Response

Thank you for your encouraging comments and helpful suggestions.

  1. Numerals instead of words: I apologise for the ‘sloppy’ use of numerals rather that words according to the convention and have checked the manuscript carefully to correct this.
  2. Viruses as cause: This was not our intention, and we are of the same opinion as the reviewer so have checked through and removed ‘cause’ and softened the wording as recommended. We prefer the word ‘trigger’ or ‘initiator’.
  3. Data conducted by the authors: We have taken excerpts of published data and teased out observations and interpretations from more comprehensive studies to illustrate the point of individual patient variation. But a ‘limitations’ statement is a good suggestion and I have added that to reflect it is ‘observation from multiple studies’ rather than a ‘definitive designed study.

These are limited observations derived from within a larger study, but collectively, with the further observations shown in the separate studies of Figure 2B & 2C, they reinforce the conclusions that there is individual patient variation within a very similar phenotype of ME/CFS. “It indicates to us more individual patient studies are warranted and might explain why we are seeing so many apparent phenotypes.

  1. Line 1008: ‘first person’ -clarified corrected. Although it reflected mainly the corresponding author’s opinion the other authors concurred, and I have modified it to reflect the collective view.
  2. Institutional Review Board Statement?

I understand the confusion. Table 1 is from a thesis that may not yet be freely available in the public domain and so I wanted to record the ethics approvals for that study. I have slightly modified the language.

  • We have checked the graphics in the submitted manuscript. The Figure or parts (Figure 2) were originally made to the specifications and have transferred well into the word document we submitted but do appear less sharp in the pdf. A misplaced hyphen has been removed from one Figure.
  • Typos /formatting Four authors independently have now carefully checked through and removed unnecessary hyphens, checked for typos, and have done a word-by-word check, including checking the formatting. We appreciate the detailed evaluation to improve the manuscript.

Reviewer 3 Report

This is an excellent paper.

I would suggest that the authors review and consider including the following:  

Friedman KJ, Murovska M, Phelby DFH, Zalewski P . -  “Our Evolving Understanding of ME/CFS” Medicina, 2021  - https://www.mdpi.com/1648-9144/57/3/200\

https://www.mdpi.com/journal/healthcare/topical_collections/PAPIS

 In 2021, Friedman and co-authors discuss the importance of classifying Long COVID, ME/CFS and other similar chronic conditions as Post Active Phase of Infection Syndromes or PAPIS for the advancement of research and clinical care. Currently, a topical collection in the MDPI publication Healthcare is dedicated to this concern.

Author Response

Thank you for your encouraging comment, and suggestion alerting us to this paper in Medicina. I am happy to add this comment and reference this work.